# Application of an Ultra-Low-Cost Passive Sampler for Light-Absorbing Carbon in Mongolia

**DOI:** 10.3390/s23218977

**Published:** 2023-11-05

**Authors:** Bujin Bekbulat, Pratyush Agrawal, Ryan W. Allen, Michael Baum, Buyantushig Boldbaatar, Lara P. Clark, Jargalsaikhan Galsuren, Perry Hystad, Christian L’Orange, Sreekanth Vakacherla, John Volckens, Julian D. Marshall

**Affiliations:** 1Department of Civil and Environmental Engineering, University of Washington, Seattle, WA 98195, USA; bujinb@uw.edu (B.B.); lpclark@uw.edu (L.P.C.); 2Center for Study of Science, Technology & Policy, Bengaluru 560095, Karnataka, India; pratyush.agrawal@cstep.in (P.A.); sree.hcu@gmail.com (S.V.); 3Department of Health Sciences, Simon Fraser University, Burnaby, BC V5A 1S6, Canada; allenr@sfu.ca; 4Axon Engineering LLC, Bellevue, WA 98006, USA; mebaum@axonengineering.com; 5School of Public Health, Mongolian National University of Medical Sciences, Ulaanbaatar 14210, Mongolia; buyantushig27@gmail.com (B.B.); jargalsaikhan@mnums.edu.mn (J.G.); 6Department of Public Health and Human Sciences, Oregon State University, Corvallis, OR 97331, USA; perry.hystad@oregonstate.edu; 7Department of Mechanical Engineering, Colorado State University, Fort Collins, CO 80523, USA; christian.lorange@colostate.edu (C.L.); john.volckens@colostate.edu (J.V.)

**Keywords:** low-cost sensor, light-absorbing carbon, household air pollution, indoor air pollution, passive sampler, elemental carbon

## Abstract

Low-cost, long-term measures of air pollution concentrations are often needed for epidemiological studies and policy analyses of household air pollution. The Washington passive sampler (WPS), an ultra-low-cost method for measuring the long-term average levels of light-absorbing carbon (LAC) air pollution, uses digital images to measure the changes in the reflectance of a passively exposed paper filter. A prior publication on WPS reported high precision and reproducibility. Here, we deployed three methods to each of 10 households in Ulaanbaatar, Mongolia: one PurpleAir for PM_2.5_; two ultrasonic personal aerosol samplers (UPAS) with quartz filters for the thermal-optical analysis of elemental carbon (EC); and two WPS for LAC. We compared multiple rounds of 4-week-average measurements. The analyses calibrating the LAC to the elemental carbon measurement suggest that 1 µg of EC/m^3^ corresponds to 62 PI/month (R^2^ = 0.83). The EC-LAC calibration curve indicates an accuracy (root-mean-square error) of 3.1 µg of EC/m^3^, or ~21% of the average elemental carbon concentration. The RMSE values observed here for the WPS are comparable to the reported accuracy levels for other methods, including reference methods. Based on the precision and accuracy results shown here, as well as the increased simplicity of deployment, the WPS may merit further consideration for studying air quality in homes that use solid fuels.

## 1. Introduction

Household air pollution is a major risk factor for death and disease, annually responsible for 3.5 million premature deaths and 92 million disability-adjusted life years (DALYs) [1,2,3,4,5]. Most premature deaths occur in low-income countries, where households have relatively higher exposure to air pollution (e.g., PM_2.5_, black and brown carbon) owing to household combustion of solid fuels for cooking and heating [6,7,8,9].

Two major challenges associated with studying indoor air pollution are cost and logistics [10,11,12]. Air pollution samplers can cost from a few hundreds to several thousands of US dollars depending on factors such as their accuracy, efficiency, robustness, and user friendliness [13,14,15]. Deployments of such samplers typically require detailed logistic planning associated with deploying and collecting equipment, such as charging equipment between deployments; coordinating visits with residents; ensuring safe and secure locations; and having access to laboratory facilities.

Passive samplers can lower costs and simplify logistics associated with in situ air quality measurements; the Washington passive sampler (WPS) aims to do so for measuring the long-term average levels of light-absorbing carbon (LAC) air pollution [16]. “LAC” refers to the carbon components of fine particulate matter (PM_2.5_), such as black and brown carbon, which strongly absorb visible light (wavelengths: 400–700 nanometers) [17]. The WPS is comparatively ultra-low-cost (~USD 5-15) and logistically straightforward to use (no electricity, expensive lab equipment, or extensive maintenance required). The WPS uses digital images to measure the change in the reflectance of a passively exposed paper filter.

Here, we build on a previous study that tested the reproducibility and precision of the WPS in 20 households in Hyderabad, India [16]. The present study aimed to determine the accuracy and uncertainty of the WPS by co-locating it with (A) quartz filters for elemental carbon (EC) analysis using an ultrasonic personal aerosol sampler (UPAS, Access Sensors Technologies, Fort Collins, CO, USA), and (B) the PurpleAir sensor (PurpleAir, Draper, UT, USA) [18,19]. This investigation is the first to compare results between the WPS and other methods, including comparison against a thermal-optical analysis, which is the gold-standard measurement method for elemental carbon.

## 2. Materials and Methods

In this section, we outline the methodology employed to intercompare three distinct measurement methods: the Washington passive sampler (WPS), a reference method utilizing elemental carbon thermal-optical analysis, and the PurpleAir sensor.

### 2.1. The Washington Passive Sampler (WPS)

As stated above, the WPS is an ultra-low-cost passive sampler for light-absorbing carbon. As deployed in the field, it consists of a downward-facing cellulose filter (Whatman 1002110 qualitative circle cellulose filter paper, GE Healthcare, Chicago, IL, USA), a filter holder, and a protective case for deployment (Appendix A) [17]. Before and after deployment, the cellulose filter was photographed in a lightbox using a Basler acA3800-14 um monochrome camera (Basler acA3800-14um, Basler AG, Ahrensburg, Germany). Each photo in the lightbox was taken of two filters: the sample and a field blank. The lightbox was lit by light-emitting diode (LED) strips located on the lid, ensuring uniform lighting conditions; trays for the sample and the blank were in a fixed position, immediately below the LEDs. The lightbox was sealed to eliminate any outside light. As described below (Section 3.2), the blank filters were used to correct for potential variance in the light intensity between the pre- and post-deployment photos.

Image analysis was conducted using MATLAB’s *imread* function (MATLAB and Statistics Toolbox Release 2023A, MathWorks, Inc., Natick, MA, USA) to measure the pixel intensity (PI), which serves as a metric for image “blackness”. A PI value of 65,535 represents the whitest or lightest point, while a PI of 0 corresponds to the darkest point in the image. The difference in the PI (ΔPI) between the post-exposure and pre-exposure images serves as a quantification of the reflectance change, attributed to the deposition of light-absorbing carbon on the filter during deployment. For ease of interpretation, we consider the change in PI to be positive when the filter darkens. Consequently, all ΔPI values presented in this study utilized the MATLAB result multiplied by negative one.

The average cost of each WPS measurement is approximately USD 5–15, dependent on the deployment details. The WPS itself does not require electricity or lab equipment and is relatively easy to assemble, deploy, and maintain. For further information about the design and application of WPS and the light box, please see Clark et al. [17].

Each cellulose filter within the WPS was subjected to photography both before and after each deployment period to assess the change in pixel intensity (ΔPI), denoting “blackness”, during that specific period. Subsequently, following each deployment period, the WPS devices were redeployed in households to accumulate further darkening, alongside freshly exposed WPS units. To investigate the impact of filter loading, we progressively increased the number of WPS devices in each sampling period. The total number of WPS devices within each household during each period is detailed as follows: two WPS units in period 1, four WPS units (comprising the original two and two newly deployed units) in period 2, six WPS units (including two from period 1, two from period 2, and two newly deployed units) in period 3, and eight WPS units in period 4 (refer to Appendix A). Additionally, for quality assurance, a total of 10 field blanks were photographed and stored within a vacuum environment during each deployment. These field blanks were matched with sample sets from individual households to correct for potential issues, such as fluctuations in camera performance or lighting conditions. For example, if both the sample and the blank exhibited darkening, we employed the change in the pixel intensity of the blank to correct the sample’s change in pixel intensity (see Section 3.2 for results related to field blanks).

### 2.2. Reference Method

To determine the accuracy of the WPS, we compared the changes in the reflectance of the WPS against a “gold-standard” reference method, in this case, elemental carbon (EC) and organic carbon (OC) aerosol analysis. The EC/OC aerosol analysis method is a thermal-optical analysis, which leverages differences in the thermal refractivity of elemental and organic carbon to separate and quantify these components on a quartz-fiber filter [20]. There are several EC/OC analysis protocols, each with a different temperature plateau, residence time, carrier gas, and/or optical charring correction. The uncertainty of each protocol may reflect the composition of the aerosol samples, the filter loading effect, and the occurrence of inorganic compounds that may enhance charring and lead to erroneous determinations [20,21,22]. For details about the NIOSH5040 protocol employed here, see Karanasiou et al., 2020 [21]. We used the elemental carbon concentration as a proxy for light-absorbing carbon, as has been carried out and suggested in several studies [22,23,24,25,26,27].

To obtain samples for elemental carbon analysis, we utilized the UPAS [18]. The UPAS is a time-integrated active sampler that can record the mass flow (MZBD001, 0.5–3.0 L/min, accurate within 5%), temperature, pressure, relative humidity, light intensity, and acceleration. To avoid the possible saturation of the filters in active samplers, such as the UPAS, we used quartz filters and a 5% duty cycle for the UPAS, meaning that the UPAS ran 5 s every 100 s. After each deployment, the UPAS samples were collected and subsequently sent to the Center for Energy Development and Health (CEDH) at Colorado State University (CSU) in Fort Collins, CO, for thermal-optical analysis. This analysis enabled us to determine the mass of elemental carbon collected on the filter. By combining this elemental carbon mass with the mass flow data recorded by the UPAS, we calculated the average elemental carbon concentration for each household during each deployment period. We compared this elemental carbon concentration against the WPS-measured average pixel intensity change recorded during the corresponding deployment period, allowing us to establish the calibration curve.

### 2.3. PurpleAir

To compare the light-absorbing carbon and elemental carbon concentrations with PM_2.5_ concentrations, we employed PurpleAir sensors, which continuously measure the PM concentration, temperature, and relative humidity. These sensors are equipped to transmit real-time data to the cloud when connected to Wi-Fi, a feature that we utilized to monitor any disruptions during deployment, such as electricity outages, and to visit the households after such disruptions.

The PM_2.5_ concentrations reported by PurpleAir sensors demonstrate a strong correlation with reference methods, like the EPA federal reference methods and federal-equivalent methods (R^2^ > 0.9). However, PurpleAir readings can occasionally overestimate or underestimate concentrations by as much as a factor of 2, primarily due to environmental variables, such as high relative humidity [28,29,30,31,32]. In our study, PurpleAir sensors were deployed without prior calibration, a common practice even though it may compromise the precision of PurpleAir measurements. Consequently, we utilized PurpleAir data for comparative purposes but excluded them from some analyses, including the establishment of the calibration curve.

### 2.4. Study Design

The study was conducted in 10 households in Ulaanbaatar, Mongolia, from December 2020 to April 2021 (~4 months). The households had a similar size and geometry, and each used government-subsidized charcoal as the main fuel source for cooking and heating. In each household, we utilized three devices (WPS, UPAS, PurpleAir; see Table 1). As described in Appendix A, the ~4 months consisted of a total of 4 deployments (i.e., sampling periods) of 21–35 days each.

### 2.5. Data Analysis

Given the study’s design, the potential maximum number of elemental carbon measurements would have been 40 duplicates (10 households over 4 deployment periods). However, this maximum data point count was not achieved due to various maintenance issues, such as UPAS shutdowns due to high temperatures, battery depletion during electricity outages, and other unexplained factors. To ensure data quality, we enforced a requirement that all UPAS measurements ran for a minimum of 24 h per deployment (i.e., the maximum sampling duration for UPAS at a 5% duty cycle was 25–42 h). As a result, the final count of elemental carbon measurements amounted to N = 21, with 10 duplicates and the rest representing single measurements, owing to the deployment issues mentioned above.

Our data analysis encompasses four main components:Precision and reproducibility of WPS and UPAS: We utilized the intraclass correlation coefficient (ICC) to determine the same-method agreement between paired duplicate WPS samples and (separately) paired duplicate UPAS samples. We also compared the precision of both methods [33]. The ICC measures how strongly the duplicate samples resemble each other; ICC = 1 means that the duplicate samples perfectly match, and therefore, the precision of the sampler is perfect/infinite. The ICC is more appropriate than R^2^ for understanding the consistency of duplicate measurements because the paired duplicate measurements are mathematically equivalent. (In contrast, R^2^ is used when the pairs have differentiation: one measurement is necessarily “x”, and the other is necessarily “y”.)Comparing the WPS against the gold standard: In this step, we assessed the performance of the WPS against the gold-standard method and established a calibration curve for the light-absorbing carbon relative to the elemental carbon. Deming regression (*deming* package in R (R version 4.1.2, R Foundation for Statistical Computing, Vienna, Austria) was used to derive the calibration curve because of the uncertainties in both the elemental carbon and the light-absorbing carbon measurements. The accuracy of the WPS was then computed as the root-mean-square error (RMSE) between the observed change in reflectance (utilizing the gold-standard method) and the predicted change in reflectance (using the WPS with the empirically determined calibration curve).Correlations across methods: This component involved examining the correlations among all three measurement methods for each household during each deployment period.Comparing WPS measurements across deployments: This component aimed to investigate potential filter-loading effects by comparing the WPS measurements across different deployment periods (i.e., across duplicate samples made using different ages of the filter and filter paper).

## 3. Results

### 3.1. Measurement Completeness

The measurement completeness, which represents the percentage of successfully executed samples out of the intended samples, was notably higher for the Washington passive sampler (WPS), with a rate of 80% (32 out of 40 intended samples) compared to 55% for the ultrasonic personal aerosol sampler (UPAS) (21 out of 40 intended samples). When focusing on duplicate-sample completeness, i.e., the percentage of intended paired duplicates that were successfully obtained, the WPS achieved a rate of 70% (28 out of 40 intended samples), while the UPAS showed a lower rate of 25% (10 out of 40 intended samples). These results underline the reliability and lower failure rate of the WPS in comparison to the UPAS, particularly during longer measurement periods.

### 3.2. Precision of WPS and UPAS

Figure 1 reports duplicate samples collected by the WPS, and (separately) the UPAS. Both methods demonstrated relatively good self-agreement. As measured by the intraclass correlation coefficient (ICC), the precision was higher for the UPAS (ICC = 0.96) than the WPS (ICC = 0.88).

### 3.3. Field Blank and Calibration Curve

The mean change in the reflectance change for the 10 field blanks was 77 (standard deviation of 20) PI per month. Before establishing the calibration curve, we adjusted the change in the reflectance for the exposed filters by subtracting the mean change in the field blanks. The Deming regression analyses to calibrate the light-absorbing carbon to the elemental carbon measurement indicate that, on average, 1 µg of elemental carbon (EC) per cubic meter corresponds to 62 PI per month (as shown in Figure 2). Applying that conversion to all WPS measurements to predict the elemental carbon concentrations from the passive light-absorbing carbon measurements, for a one-month measurement, the root-mean-square error is 3.1 µg/m^3^ EC. That value corresponds to ~21% of the average elemental carbon concentration.

In summary, the average concentrations were 180 µg/m^3^ for PM_2.5_ (PurpleAir, uncalibrated) and 14.1 µg/m^3^ for elemental carbon (measured by the UPAS) from the 21 samples collected across various households during each deployment period. The light-absorbing carbon measurements, expressed in their original, uncorrected units, reflect the rate of change in the filter color, denoted in units of change in pixel intensity per month, which depends on the deposition of the light-absorbing carbon over time. The average value for the change in pixel intensity (ΔPI) was 1052 PI per month for all 32 samples collected, and 952 for the samples collected from the same subset of households where elemental carbon samples were successfully obtained. Using the calibration value (1 µg of EC per m^3^ corresponds to 62 PI per month), 1052 PI per month would correspond to an average of 17.0 µg/m^3^. Alternatively, when using the calibration curve shown in Figure 2 (y = 62.1x + 75.1), 1052 PI per month would correspond to 15.7 µg/m^3^. Those two values suggest that if the number of successful samples for elemental carbon analysis were 32 instead of 21, the average concentration would have been greater than 14.1 µg/m^3^.

### 3.4. PurpleAir

In our comparison of the PM_2.5_ concentration in households with both light-absorbing carbon and elemental carbon, the results, as depicted in Figure 3, indicate that both light-absorbing carbon and elemental carbon exhibited strong correlations with PM_2.5_. Specifically, the UPAS (measuring elemental carbon) demonstrated a slightly stronger correlation with PM_2.5_, with an R^2^ value of 0.88, in comparison to the WPS (measuring light-absorbing carbon), with an R^2^ value of 0.77. This observation likely reflects the diverse sources and chemical composition of PM_2.5_, as well as, and perhaps more significantly, the slightly lower precision of the WPS compared to the UPAS in this context.

### 3.5. Darkening Rate of Fresh and Aged Filters

Here, we compare the darkening rates of aged (exposed for more than 4 weeks) and fresh (exposed for 4 weeks) filters (Figure 4). The darkening rates were similar but not identical (ICC = 0.90); the 95% CI on the best-fit line (the shaded blue region in Figure 4) includes the 1:1 line. There is moderate evidence of a modest difference in the filter darkening rate for the aged and fresh filters (slope = 1.06 ± 0.05). This “filter loading” effect is well-documented for micro-aethalometers (Good et al., 2017) [34] which use similar principles to the WPS.

## 4. Discussion

The objective of this study was to assess the performance and uncertainty of the Washington passive sampler (WPS), an innovative, cost-effective passive sampler designed for measuring light-absorbing carbon. The precision and accuracy of the WPS were determined through the deployment of multiple duplicate samples and co-location with a PM2.5 proxy method (PurpleAir), as well as duplicates of the “gold-standard” reference method (UPAS with elemental carbon analysis). Notably, unlike the active PurpleAir and the gold-standard UPAS method, the WPS is a passive sampler. The study’s findings indicate that changes in the reflectance measured by the WPS can effectively predict the long-term average elemental carbon concentration with a relatively good level of accuracy, as indicated by a root-mean-square error (RMSE) of 21%.

The reported accuracy of the Washington passive sampler (WPS) with an RMSE of 21% is consistent with the accuracy levels reported for other methods used to measure black carbon or elemental carbon [35,36,37,38,39,40,41,42,43]. For instance, low-cost black carbon samplers available on the market exhibited similar accuracy, such as an RMSE of 25% reported for ABCD, and approximately 10% for another image-based reflectance method [12,36].

Chiappini et al. [22] conducted a study involving three sets of duplicated co-located samples, each analyzed using three different protocols for elemental carbon (EC) analysis. They reported overall uncertainties of 14%, 39%, and 20% for the respective sets (averaging 24%) for samples #1, #2, and #3, indicating that elemental carbon analysis can have variable uncertainties across different protocols. Several inter-laboratory comparison studies have also reported relative standard deviations of 6–26% for elemental carbon analysis across various widely used thermal-optical analysis protocols in different laboratories (see Appendix A) [21,22,37,38,39,40,41,42,43]. It is important to note that the RMSE, a measure of accuracy, is the combination of the standard deviation (a measure of precision) and absolute bias [44]. Therefore, the RMSE, as reported here for the WPS, is expected to be higher than the standard deviation. The accuracy of the WPS aligns with the levels reported in the literature for other methods, including the gold-standard reference methods, underscoring its reliability and effectiveness for measuring light-absorbing carbon.

The correlation between the PM_2.5_ and light-absorbing carbon, indicated by an R^2^ value of 0.77, was slightly lower than the correlation between the PM_2.5_ and elemental carbon, which had an R^2^ value of 0.88. This implies that the relationship between the PM_2.5_ and elemental carbon is slightly stronger. Additionally, the darkening rates of the aged and fresh filters exhibited good correlations, as reflected by an intraclass correlation coefficient (ICC) of 0.90. The 95% confidence interval (CI) region of the best-fit line includes the 1:1 line, and the slope of the line for the aged versus fresh filters was 1.06 (±0.05). The minor difference observed, with the darkening rates being slightly lower for the aged filters (i.e., darker) compared to the fresh filters (i.e., less dark), aligns with the well-documented filter-loading effect observed for micro-aethalometers [34].

A recent study by Jeronimo et al. [36] also developed a low-cost method for estimating the concentration of black carbon using a digital camera. They compared the image-based reflectance method to several existing reference methods, including thermal-optical analysis, and found a good correlation with a normalized RMSE of less than 10% for all comparisons. Both studies (ours; Jeronimo et al. [36]) developed a digital image-based method and compared it against a gold-standard method; the main differences include the sampling method (Jeronimo et al. [36] employed active sampling instead of passive samplers), filter type (Jeronimo et al. [36] used more expensive PTFE filters, whereas our method utilized cellulose filters), and sampling time (in Jeronimo et al.’s study, the sampling duration ranged from 24–48 h, whereas our study extended over 21–35 days). Nevertheless, the results of Jeronimo et al.’s study offer consistent and reassuring evidence that an image-based reflectance method can deliver accurate estimates.

Relative to the gold-standard method (thermal-optical analysis of active filters), the WPS offers important advantages in scalability, ease of use, utility, lower measurement failure rate, and cost. Most studies of indoor air pollution effects on human health are based on exposure data collected for 24–48 h, owing to the high cost of measurement devices and the logistics of measurements [28,43]. Thus, the results of these studies rely on short-term average concentrations of black carbon, which can be different with changing environments and household behavior. In some cases, studies with filters employ multiple visits (e.g., three samples of 24 h each, for an 18-month period), thus providing a small number of snapshots. In contrast, the WPS can be deployed for months or potentially years without requiring extensive maintenance, and it allows for the collection of long-term levels of LAC concentrations. For many investigations, the lower cost of measurement and the greater ease of use of the WPS relative to the gold-standard measurements, combined with the opportunity of using the WPS to obtain long-term rather than short-term averages, may offer important advantages for exposure and health studies in households that use solid fuels.

Recent studies comparing low-cost methods on the market to reference methods all suggest that the readings of low-cost sensors are sensitive to environmental factors, such as the relative humidity, and performances against reference monitors change with the weather [42,43,45,46]. Light-absorbing carbon estimates using image-based methods for reflectance may be sensitive to the site, season, pollution source (e.g., fuel type), and/or reference method selected for calibration. Further testing needs to be carried out to determine the (i) correction factor under different environmental conditions, (ii) the calibration curve of light-absorbing carbon against PM_2.5_, and (iii) the upper and lower thresholds of the detection limit in terms of concentration and time.

## 5. Conclusions

This study evaluated the Washington passive sampler (WPS) as a novel, ultra-low-cost sensor for assessing the levels of light-absorbing carbon. A prior field campaign [16] investigated the precision of this method; the present campaign aimed to investigate its accuracy. To do so, we compared duplicate samples of the WPS against co-located duplicate samples using a gold-standard method, thermal-optical analysis.

Our results indicate that the root-mean-square error (RMSE) of the WPS was 21%, which is comparable to the literature-reported values for other methods, including gold-standard methods. This level of accuracy is surprisingly high, given the ultra-low cost and the ease of use of this method. The two main differences between the WPS and the gold-standard measurements are the measurement type (image analysis versus chemical analysis) and passive versus active samples (i.e., without versus with a pump).

The strong correlation with elemental carbon (R^2^ = 0.88) and PM_2.5_ (R^2^ = 0.77), along with the consistency observed in the darkening rates of aged and fresh filters (i.e., no strong filter-loading effect), further corroborates the WPS’s reliability. The WPS offers advantages in terms of its scalability, ease of use, cost-effectiveness, and a prolonged deployment approach, which enables collecting data on long-term levels of light-absorbing carbon pollution. In our study, the percentage of successful samples collected over the intended sample number was notably higher for the WPS (80%) compared to the UPAS (55%) in total. When considering duplicate samples, the difference remained substantial, with the WPS achieving a success rate of 70% versus the UPAS’s 25%. While these challenges encountered during data collection may be unique to our study, issues such as unintentional shutdowns, due to factors like high temperatures, battery depletion from extended power outages, and unexplained operational disruptions, could be encountered universally in field conditions, especially in regions like rural India. Based on our findings, the WPS appears to be a well-suited choice for long-term studies in rural locations, offering a reliable, practical, and inexpensive solution for measuring light-absorbing carbon pollution.

This paper represents only the second publication regarding the WPS [16]; additional testing of its robustness and methods for deployment would be helpful. For example, further research on the sensitivity of the WPS to environmental factors, the detection limit, and the calibration curve against elemental carbon and PM_2.5_ would usefully shed light on the WPS’s reliability across diverse environmental contexts.

## Figures and Tables

**Figure 1 sensors-23-08977-f001:**
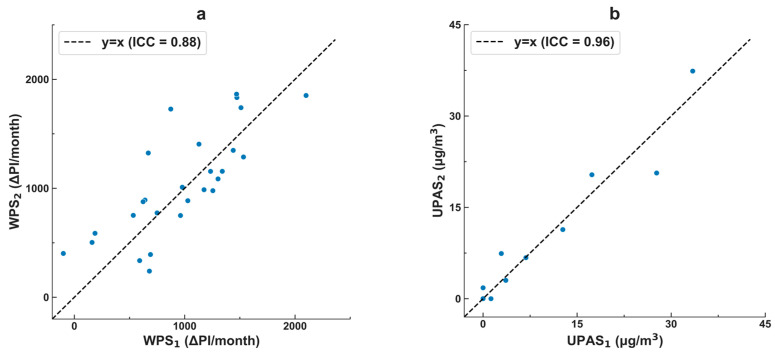
Correlation between duplicate samples: (**a**) Washington passive samplers (WPS), which measure the passive deposition of light-absorbing carbon (LAC), and (**b**) filter samples collected via an ultrasonic personal aerosol sampler (UPAS) and thermo-optically analyzed for elemental carbon (EC). ICC is the intraclass correlation coefficient, a measure of how strongly the duplicate measurements resemble each other. The elemental carbon collected by the UPAS measurement is considered here the “gold standard”; the light-absorbing carbon collected by the WPS measurement is the new method investigated here. The plots show that the duplicate WPS and UPAS each exhibited strong within-method agreement (ICC > 0.88), but precision was higher for the UPAS than the WPS. There are more measurements shown for the WPS than for the UPAS (n = 28 vs. n = 10), reflecting that the WPS is simpler and less failure-prone than the UPAS; the percentages of intended duplicate samples that were successfully obtained were 70% for the WPS and 25% for the UPAS.

**Figure 2 sensors-23-08977-f002:**
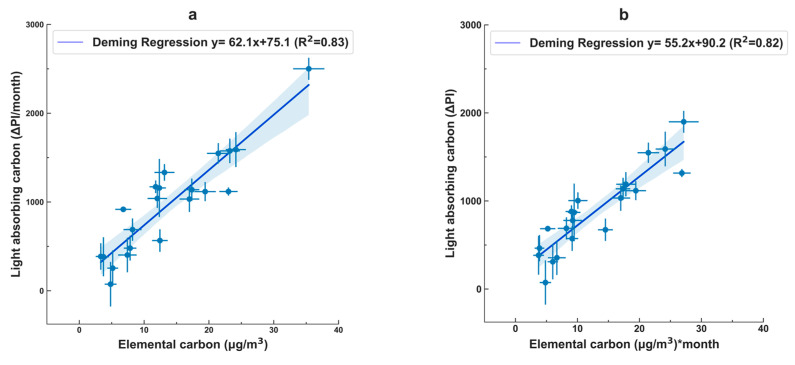
Correlation between the “gold standard” (*x*-axis: EC) and the WPS (*y*-axis: LAC), displayed using the native units of (**a**) the elemental carbon measurements and (**b**) the light-absorbing carbon measurements. (The two plots represent the same data, displayed in different units; the plots are similar but not identical because the deployment duration differed by the sampling period.) The y-error bars represent the range of the duplicate WPS, and the x-error bars represent the uncertainties from the elemental carbon analysis. Deming regression was utilized (R, Deming package), reflecting that both measurements have uncertainties.

**Figure 3 sensors-23-08977-f003:**
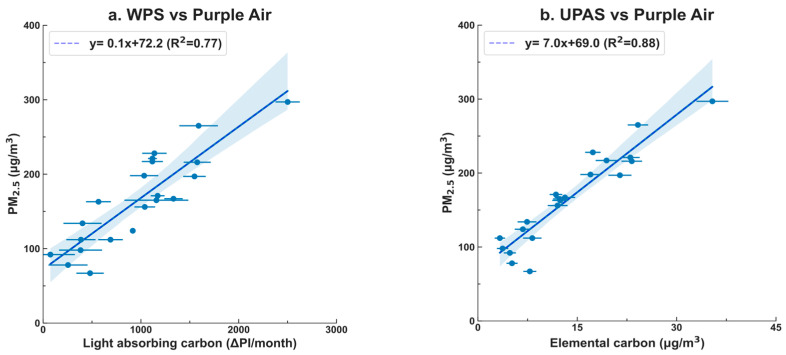
Correlation between PurpleAir (uncalibrated) with WPS (**a**) and elemental carbon analysis (**b**). The x-value error bars represent the difference between the duplicate WPS results (**a**) and the elemental carbon analysis uncertainties reported by the lab (**b**).

**Figure 4 sensors-23-08977-f004:**
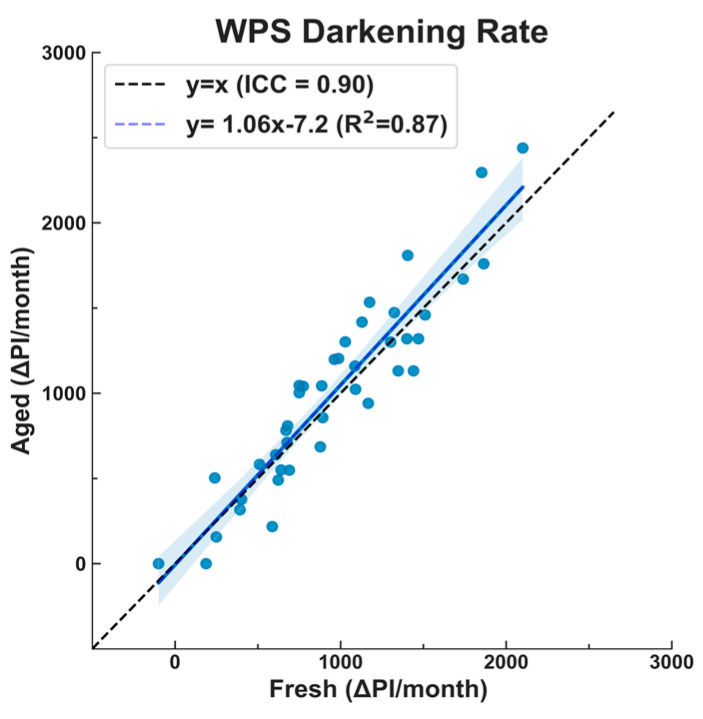
Darkening rate difference between fresh and aged filters. Fresh filters were photographed before deployment and ~4 weeks after exposure in the household; aged filters were photographed after being exposed for a certain period, then photographed after ~4 more weeks of exposure, i.e., aged filters represent the delta pixel intensity of already exposed filters after more exposure. The results here are similar to the “filter loading” effect, which is well-documented for micro-aethalometers.

**Table 1 sensors-23-08977-t001:** Deployment details.

Device	Design	Sampling Type	Filter/Sensor Type	Measuring Species	Comments
WPS	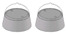	Passive, time integrated. Ultra-low cost.	Cellulose(Whatman)	Light absorbingcarbon (LAC)	Each WPS was photographed before and after deployment.
UPAS	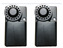	Active, time integrated. Reference method.	Quartz (37 mm)	Elemental carbon (EC)	Each UPAS was connected to electricity and continuously sampled at a 5% duty cycle during the deployment period.
PurpleAir	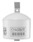	Active, real-time.Medium-cost.	Continuous sensors ^1^	PM_2.5_	Electricity outages were determined with PurpleAir Map.

^1^ PurpleAir (uncalibrated) measured indoor PM_2.5_ concentration using PMS5003 laser particle counters and reported real-time concentration to PurpleAir Map if connected to Wi-Fi; this feature was used to log electricity outages.

## Data Availability

Data available on request.

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
