# Peer review of "Application of an Ultra-Low-Cost Passive Sampler for Light-Absorbing Carbon in Mongolia"

_sensors, 2023, doi:10.3390/s23218977_

Round 1

Reviewer 1 Report

Comments and Suggestions for Authors

The article is quite interesting and useful. Therefore, I recommend it for publication. However, the description of the methodology needs to be reworked. It lacks detail. Also the information in the results are rather limited. The authors should rework that as well. They need to give more arguments that can be learned from the figures why they think their method is reliable. 

-          Paragraph 2.1: Would the title “Ultra-low-cost passive sampler” not been better? After all, this is the main topic of the article.

-          Paragraph 2.1: it is stated that the filter is photographed before and after exposure. What happens when the illumination in the lightbox is not constant over time? Is it not better to photograph the filter with a blanco reference? Information of line 104-105 belong to paragraph 2.1. Give the formula how you calculated the blackness. If you would divide grey value after divided by gray value before exposure, I would expect a value between 0 and 1. The article reports values higher than 1000. This requires explanation. Do you place a closed blank in the houses or are they stored in vacuum in the laboratory? Please explain?

-          Paragraph 2.1: It is not clear what the procedure is to determine the blackness. Do you convert colour images to black and white? How do you determined the average grey value?

-          Paragraph 2.2: There are many abbreviations in this paragraph and this makes it hard to read.

-          Paragraph 2.2: EC aerosol analysis is a thermal-optical analysis that involves separation 80 and quantification of EC sampled on a quartz-fiber filter [20]. The term “separation” is not entirely clear to me. Doe sit refer to the removal of carbon from the filter or to a chromatographic technique?

-          Paragraph 2.2, line 84: What is the brand name of the active sampler What is the total amount of air that has been sampled (or what is the sampling time?)

-          Table I: Mention what is the low-cost method and what is the gold standard in the table. The PurpleAir has not been mentioned in paragraph 2.1 or 2.2 so it is not clear to me what its role is. It is only later in the article that it becomes clear that 3 different methods are compared with each other. Make that comparison clear in the materials and methods section.

-          The paragraph that starts at line 101 below Table I can be better split up over paragraph 2.1 and paragraph 2.2. Only the information about the relationship between low-cost and gold standard should remain here. The analysis of relationships is given in line 123. I recommend to put all information about relationships together. I have the impression that different types of relationships have been studied. Make this more explicit. Given this section a title so that the the blocks Low-cost passive sampler, gold standard, relationships becomes more obvious. Emphasize how ypu determine the reliability of the low-cost method.

-          I am wondering if it would not be easier to summarize all measurements in all households with a table. Every row in a measuring campaign. In the columns you can give the period, number of the house being measured, method used, etc. You can also number the measurements so that you can refer to a specific measurement if needed.

-          Figure 1: Replace the abbreviations with full names (e.g., ultra-low cost method by using the blackness, gold standard by using elemental carbon). The low cost method is not measuring the LAC but the blackness of the filter. Is the variation in both methods caused by errors in the measurements or by a heterogeneous distribution of black carbon in the houses analysed? Elaborate what you see in the graphs.

-          Figure 2: Avoid the abbreviations

-          Results: In several sections the main text in the results is smaller than the caption below the figures. Try to elaborate what you have learned from these figures. For example, how should figure 1 convince me – the reader- that the low-cost method is reliable.

-          What is meant by aged filter? A filter that has some age that has a yellowish hue? I don’t think this type of blank has been mentioned in paragraph 2.1

Comments on the Quality of English Language

-          The article contains so many abbreviations, that it is hard to read. For example, in the figures it is not needed to use the abbreviations.

-          Line 50: Deployments … require

Reviewer 2 Report

Comments and Suggestions for Authors

Dear editor journal of Sensors

The manuscript titled" Application of an ultra-low-cost passive sampler for light-absorbing carbon in Mongolia" determine the accuracy and uncertainty of WPS, the manuscript structure need to be improved in term of clarity and discussion of results and conclusion.

-Abstract: PM2.5, RMSE, …; When you use an abbreviation in both the abstract and the text, define it in both places upon first use (UPAS in the introduction).

-Method: The total number of samples clearly be stated.

-Normally, comparisons are made based on statistical tests, but in this study, only correlation is reported, so on what basis is the comparison made?

- The caption of the figures should be separated from the result

-Discussion of the study should be improved

-Please consider the conclusion section of the study.

Round 2

Reviewer 1 Report

Comments and Suggestions for Authors

The article has been improved a lot. Congratulations to the authors. However, I still have many important remarks that need to be tackled before it can be published:

·        Concerning the photograph of the filters, it is still not entirely clear how this was done. Is this a setup with a light source at 45° right and left from the sample? An important issue is that lamps do not generate a constant intensity over time. It is not clear if this has been considered and how the correction was done. Is the result always intensity of the sample filter divided by the intensity of the blanco? Later in the results part, a subtraction has been suggested. Be more explicit.

·        Line 131: The sentence below Table 1 seems to be quite important to understand the article but the amount of information is too limited. I suppose that the different methods have been used in parallel and the that measurements of WPS and PurpleAir have been plotted in function of the results of the gold standard. Is that correct. Please describe how you did the calibration in detail.

·        Section 3.1: A section of 1 sentence is rather strange. Explain what I have to see in the graph. Explain what the points in Fig. 1 mean. What is the meaning of ICC. IS it the coefficient of determination of the linear regression? Please explain. The conclusion that the low-cost method can be applied is not stated. If I understood it well, each location has been analysed by 2 methods in parallel. Is that correct? This is elementary information that needs to be included in the article.

·        The font size of Fig. 1 is too small. It is hard to read the texts. Make them bigger

·        Section 3.2: What I do not understand is that the first paragraph is about the analysis and then followed by a paragraph about the blanco. Is it not more logical to reverse the order of these paragraphs. And what is PA. This abbreviation is not mentioned in Table 1.

·        Section 3.3: You need to explain what I learn from the graph. There is no information how reliable the PurpleAir is. In addition, I would not immediately expect a good correlation between particulate matter and soot. Why is that?

·        Section 3.4: It is not clear to me what is darkening. Do you mean that the blanco and the filter after being exposed to ambient air continues to darken? Since this is a methodological question, it should be better to start the results part with the darkening. Is the darkening due to yellowing of the paper caused by UV-radiation or due to a slow absorption of soot? Explain.

·        Would it be possible to use full names instead of all the abbreviations. It makes the reading of your valuable article quite hard.

Comments on the Quality of English Language

Please check your English. Let it correct by chatGPT. You will get a more fluent text. Ask chatgpt the following "Can you correct the following text?"

Reviewer 2 Report

Comments and Suggestions for Authors

Dear editor journal of Sensor

The authors have addressed my comments

Comments on the Quality of English Language

Minor editing of English language required

Author Response

Thank you for your comments.